# Interaction between CO$_2$-consuming autotrophy and CO$_2$-producing heterotrophy in non-axenic phototrophic biofilms

Patrick Ronan[1], Otini Kroukamp[1], Steven N. Liss[1,2], Gideon Wolfaardt[1,2]*

**1** Department of Chemistry and Biology, Ryerson University, Toronto, ON, Canada, **2** Department of Microbiology, Stellenbosch University, Stellenbosch, South Africa

* gmw@sun.ac.za

**Data Availability Statement:** All relevant data are within the paper.

**Funding:** This work was supported by the Natural Sciences and Engineering Research Council of

## Abstract

As the effects of climate change become increasingly evident, the need for effective CO$_2$ management is clear. Microalgae are well-suited for CO$_2$ sequestration, given their ability to rapidly uptake and fix CO$_2$. They also readily assimilate inorganic nutrients and produce a biomass with inherent commercial value, leading to a paradigm in which CO$_2$-sequestration, enhanced wastewater treatment, and biomass generation could be effectively combined. Natural non-axenic phototrophic cultures comprising both autotrophic and heterotrophic fractions are particularly attractive in this endeavour, given their increased robustness and innate O$_2$-CO$_2$ exchange. In this study, the interplay between CO$_2$-consuming autotrophy and CO$_2$-producing heterotrophy in a non-axenic phototrophic biofilm was examined. When the biofilm was cultivated under autotrophic conditions (i.e. no organic carbon), it grew autotrophically and exhibited CO$_2$ uptake. After amending its growth medium with organic carbon (0.25 g/L glucose and 0.28 g/L sodium acetate), the biofilm rapidly toggled from net-autotrophic to net-heterotrophic growth, reaching a CO$_2$ production rate of 60 μmol/h after 31 hours. When the organic carbon sources were provided at a lower concentration (0.125 g/L glucose and 0.14 g/L sodium acetate), the biofilm exhibited distinct, longitudinally discrete regions of heterotrophic and autotrophic metabolism in the proximal and distal halves of the biofilm respectively, within 4 hours of carbon amendment. Interestingly, this upstream and downstream partitioning of heterotrophic and autotrophic metabolism appeared to be reversible, as the position of these regions began to flip once the direction of medium flow (and hence nutrient availability) was reversed. The insight generated here can inform new and important research questions and contribute to efforts aimed at scaling and industrializing algal growth systems, where the ability to understand, predict, and optimize biofilm growth and activity is critical.

## Introduction

The accumulation of carbon dioxide in the atmosphere caused by the burning of fossil fuels is widely reported as being a leading cause of climate change. The effects of climate change are far-reaching and evident in the form of increasing global temperatures, melting of glacial ice,

Canada (NSERC) in the form of a Postgraduate Doctoral Scholarship (PR). https://www.nserc-crsng.gc.ca/index_eng.asp The funders had no role in study design, data collection and analysis, decision to publish, or preparation of the manuscript.

**Competing interests:** The authors have declared that no competing interests exist.

rising sea levels, and other well-documented negative impacts [1]. Strategies for effective CO$_2$ management and mitigation are therefore critical in order to minimize these destructive effects and ensure the prosperity of current and future generations.

In addition to rising atmospheric CO$_2$ levels, access to clean water is another major environmental and health concern, impacting nearly a billion people globally [2]. Even in developed regions with contemporary water treatment practices, increasing population growth and density means that existing treatment infrastructure is often strained, operating at or above capacity. As a result, untreated or undertreated water is frequently discharged to the environment, polluting watersheds and threatening nearby communities [3]. Inorganic nutrients in wastewater are especially concerning given that their accumulation in receiving waters leads to eutrophication and places ecosystems and water-users at serious risk [4, 5]. Despite being a major component of wastewater treatment, classical biological nutrient removal processes are energy-intensive and typically configured for optimal nitrogen or phosphorus removal, but not both. As a result, these processes do not always meet the desired removal efficiency [6, 7].

In view of environmental concerns, solutions which can effectively and simultaneously address the related issues of CO$_2$ sequestration and enhanced wastewater nutrient removal are urgently needed; there is growing indication that processes incorporating microalgae may contribute to such optimization [7]. These photoautotrophs are capable of rapid CO$_2$ uptake and fixation via their photosynthetic metabolism, generating the energy-rich sugars needed to fuel cellular activities. Like plants, microalgae readily assimilate inorganic nitrogen and phosphorus, meaning there is potential to utilize wastewater as a cheap and abundant algal growth medium [8–11]. As an added benefit, algal biomass has inherent commercial value, and can be used for example as a bio-fertilizer [12]. Microalgae also produce various useful compounds and nutraceuticals [13] and are known to be a potentially valuable biodiesel feedstock [14, 15]. This value-added quality of algal biomass can reduce or offset process-related costs. Although the range of allowable applications for waste-grown biomass remains somewhat restricted [16], further research and evidence-based policymaking focused on risk mitigation could lead to a paradigm in which microalgal CO$_2$-sequestration, enhanced wastewater treatment, and biomass generation may be effectively combined [7, 17, 18].

Despite the ostensible triple-benefit of such an approach, its scalability is also limited by factors like high costs and energy requirements [11]. While the use of microalgal biofilms instead of conventional suspended cultures can alleviate land footprint requirements and reduce the cost of biomass harvesting and dewatering [19], widespread adoption of algal biotechnology is also hindered by a lack of robustness [20]. This is of particular concern given the common use of axenic cultures, which contain only a single algal species. Such cultures are typically constrained to a relatively narrow range of growth conditions and are highly susceptible to contamination, leading to sub-optimal performance or culture collapse if a sterile aseptic environment is not maintained [18, 21, 22].

To overcome this challenge, there is growing focus on the use of natural, non-axenic cultures in engineered algal systems [10, 20, 21, 23, 24]. The inherent diversity and functional redundancy within such communities confers greater robustness and makes them an attractive option for use in photobioreactor systems [19, 25]. As such, a shift in our approach to algal biotechnology and biosequestration, from one focused on finding ideal species to one focused on managing the environment to select for desired species interactions, may help to maximize process efficacy and resilience. However, to effectively realize such an approach, there is a need for an improved fundamental understanding of non-axenic phototrophic biofilms, their behaviour, and associated internal interactions within engineered growth systems.

The present work aims to contribute in this regard by examining the interplay between CO$_2$-consuming autotrophy and CO$_2$-producing heterotrophy within a non-axenic

phototrophic biofilm. This question is highly relevant to integrated $CO_2$ sequestration-wastewater treatment systems in which a phototrophic biofilm may be simultaneously exposed to both inorganic and organic carbon sources. That some microalgal species can grow mixotrophically utilizing both inorganic and organic carbon, and a few are even capable of growing heterotrophically on organic carbon only [26], makes the study of autotrophic-heterotrophic interplay in phototrophic biofilms especially prudent. The insight gained through this work may contribute information for optimizing growth system design, operation, and performance.

In this study, phototrophic biofilms were grown in a $CO_2$ Sequestration Monitoring System (CSMS) [27]. Building upon the $CO_2$ Evolution Monitoring System (CEMS) described by Kroukamp and Wolfaardt [28], this system enables the real-time in situ monitoring of a phototrophic biofilm's $CO_2$ flux under varying conditions. This work was motivated by a series of hypotheses: i) that non-axenic phototrophic biofilms readily toggle between net-autotrophic and net-heterotrophic growth depending on the availability of labile organic carbon sources; ii) that in the presence of both $CO_2$ and labile organic carbon, phototrophic biofilms exhibit distinct, longitudinally discrete regions of heterotrophic and autotrophic growth with distance from reactor inflow; and iii) that this distinct longitudinal separation of autotrophic and heterotrophic metabolism is transient and reversible, based on the flow direction (and hence availability) of organic carbon nutrients.

## Materials and methods

### Growth system

Biofilms were grown in a $CO_2$ Sequestration Monitoring System (CSMS). This system, described in detail in Ronan et al. [27], enables the real-time, in situ monitoring of $CO_2$ flux in phototrophic biofilms. Biofilm reactor (BR) modules were 150 cm in length and comprised a tube-within-a-tube design (Fig 1A). Biofilms grew inside a $CO_2$ permeable silicone tube (1.57 mm ID, 2.41 mm OD, 0.41 mm wall thickness, 2013.2 Barrer permeability; VWR International, Mississauga, ON, Canada), through which liquid growth medium flowed. The silicone tube in each BR provided approximately 74 cm$^2$ of colonizable surface area and was housed within a larger diameter Tygon™ tube (4.76 mm ID, 7.94 mm OD, 1.58 mm wall thickness, E-3603 formulation; VWR International, Mississauga, ON, Canada). A $CO_2$-free sweeper gas (TOC grade, 24001980; Linde Canada Limited, Concord, ON, Canada) was channeled through the annular space created by the two tubes. In this configuration, $CO_2$ molecules could readily diffuse from the biofilm into the annular space (Fig 1B), or from the annular space into the biofilm (Fig 1C), depending on the nature of the biofilm and the prevailing $CO_2$ concentration gradient.

In this study, the CSMS consisted of three sequential BRs (Fig 2). The first, $BR_{prod}$ ($CO_2$-*producer*), received fresh growth medium and housed a pure culture heterotrophic bacterial biofilm. The purpose of this biofilm was to provide a steady source of $CO_2$ to the phototrophic biofilm being studied in the two subsequent BRs downstream. The sweeper gas passed through the annular space of $BR_{prod}$, collecting $CO_2$ produced by the heterotrophic biofilm and carrying it toward a non-dispersive infrared $CO_2$ analyzer (Analyzer 1) (LI-820; LI-COR Biosciences, Lincoln, NE, USA). The gas stream then travelled through the annular space of the two downstream biofilm reactors $BR_{cons}$ and $BR_{cons2}$ ($CO_2$ *consumer*), via a second $CO_2$ analyzer (Analyzer 2) positioned between them. Finally, the gas passed through a third $CO_2$ analyzer (Analyzer 3), placed immediately downstream of $BR_{cons2}$. Each $CO_2$ analyzer has a precision in the range of 1 ppm and was set to log $CO_2$ concentrations at one-minute intervals.

The linked $BR_{cons}$-$BR_{cons2}$ unit received its own feed of fresh growth medium and housed a phototrophic biofilm, which was inoculated by injecting 6 mL of culture into the liquid

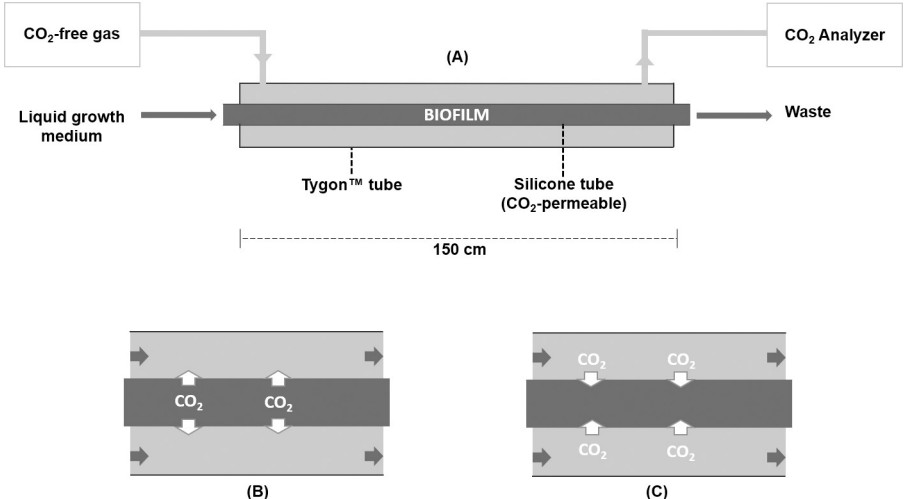

**Fig 1. CSMS biofilm reactor modules.** (A) Biofilm inoculation and growth occurs inside a highly $CO_2$-permeable silicone tube, through which liquid growth medium is pumped at a constant flow rate [27]. The comparatively $CO_2$-impermeable Tygon™ tube housing this silicone tube creates an annular space through which the sweeper gas is channeled at a constant flow rate. $CO_2$ molecules can readily diffuse in either direction across the wall of silicone tube according to the concentration gradient. (B) For a $CO_2$-producing (i.e. heterotrophic) biofilm, $CO_2$ molecules diffuse from the aqueous environment inside the silicone tube into the dry annular space. (C) When $CO_2$-laden gas is channeled into a BR containing a $CO_2$-consuming (i.e. autotrophic) biofilm, $CO_2$ molecules are pulled in the opposite direction, from the annular space into the silicone tube.

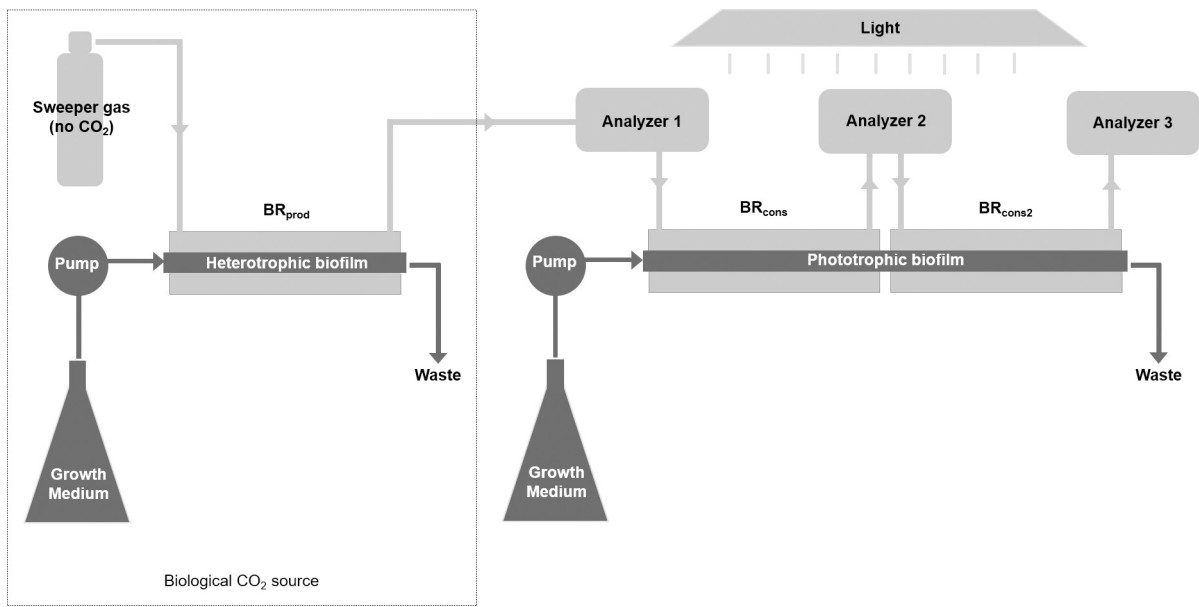

**Fig 2. Configuration of the CSMS.** The CSMS comprised three BR modules [27]. The first, $BR_{prod}$, received its own feed of fresh growth medium and housed a heterotrophic pure culture bacterial biofilm, providing a consistent source of $CO_2$. As $CO_2$ molecules diffused out of the $BR_{prod}$ silicone tube into the annular space, they were carried downstream by the sweeper gas to a $CO_2$ analyzer (Analyzer 1). The gas stream then travelled through the annular space of the linked $BR_{cons}$-$BR_{cons2}$ via a second $CO_2$ analyzer (Analyzer 2) positioned between them, before terminating at a third and final $CO_2$ analyzer (Analyzer 3). $BR_{cons}$-$BR_{cons2}$ housed the phototrophic biofilm of interest. The difference in $CO_2$ concentration logged by the analyzers provided a direct measure of $CO_2$ flux in the biofilm. Since medium flow was continuous through the linked $BR_{cons}$-$BR_{cons2}$ modules, they represent two halves of one biofilm system approximately 300 cm in length.

influent tube of $BR_{cons}$ using a sterile syringe needle. Medium flow was paused for a period of one hour following inoculation to allow for some initial cell adherence. Given that growth medium flowed continuously through $BR_{cons}$-$BR_{cons2}$, these BRs represent two halves of one biofilm system with a combined length of 300 cm.

The specific placement of the three $CO_2$ analyzers in the CSMS allowed for the direct measurement of the $CO_2$ concentration: i. entering the phototrophic $BR_{cons}$-$BR_{cons2}$ unit, ii. at the halfway point of $BR_{cons}$-$BR_{cons2}$, and iii. exiting $BR_{cons}$-$BR_{cons2}$. Therefore, at any time point it was possible to determine the $CO_2$ flux of the entire phototrophic biofilm (Eq 1), or for the proximal and distal halves of the biofilm separately (Eqs 2 and 3).

$$\text{Combined } BR_{cons}-BR_{cons2} \; CO_2 \text{ flux} = \text{Analyzer 3} - \text{Analyzer 1} \tag{1}$$

$$BR_{cons} \; CO_2 \text{ flux} = \text{Analyzer 2} - \text{Analyzer 1} \tag{2}$$

$$BR_{cons2} \; CO_2 \text{ flux} = \text{Analyzer 3} - \text{Analyzer 2} \tag{3}$$

Using the ideal gas law, the $CO_2$ concentrations measured by the analyzers were converted to rates of $CO_2$ flux and presented in units of μmol/h. Negative $CO_2$ flux values denote net $CO_2$ uptake, while positive values denote net $CO_2$ production by the biofilm.

The linked $BR_{cons}$-$BR_{cons2}$ unit was illuminated continuously via a fluorescent plant growth light (JSV2 Jump Start T5 Grow Light System; Hydrofarm Inc., Petaluma, CA, USA), resulting in a photosynthetic photon flux density (PPFD) of approximately 141.79 μmol/m²/s. Both $BR_{prod}$ and $BR_{cons}$-$BR_{cons2}$ were fed their respective growth media at a flow rate of 15 mL/h. The $BR_{cons}$-$BR_{cons2}$ retention time and dilution rate was 23.2 minutes and 2.58 h$^{-1}$, respectively. This dilution rate exceeds the maximum specific growth rates of microalgal and bacterial strains [29–31], ensuring that suspended, non-biofilm-bound cells were readily washed out the system, and hence contributed minimally to the consumption or production of $CO_2$. The sweeper gas flowed through the system at a constant flow rate of 150 mL/h.

In this study, $CO_2$ was provided to the phototrophic biofilm via a heterotrophic bacterial biofilm. At steady state, the biofilm's consistent $CO_2$ output serves to demonstrate the supply of an inexpensive and renewable $CO_2$ source to facilitate the growth and subsequent study of the phototrophic biofilm downstream. While it would also be feasible to accomplish this using a gas tank with a known or controllable $CO_2$ concentration, this approach highlights the utility in the design of the CSMS biofilm reactor modules, where the permeability of the inner silicone tube can facilitate not only the delivery of $CO_2$ to a biofilm, but also the collection and subsequent shuttling of $CO_2$ out of a biofilm. A scaled-up CSMS-based system could also feasibly integrate $CO_2$-laden industrial flue gas or fermentation gas to support algal growth and achieve $CO_2$ bio-sequestration, a notion which is garnering increasing attention [32–35]. Delivering $CO_2$ in the gas phase through a $CO_2$-permeable membrane, as is the case in the CSMS, may also reduce the extent of pre-processing required for these gas streams. Additionally, through relatively minor modifications to the system, it would be feasible to re-circulate the $CO_2$-laden gas in a closed-loop configuration in order to improve overall sequestration efficiency.

## Test cultures and growth medium

For each experiment, an overnight culture of the heterotrophic bacterium *Pseudomonas aeruginosa* (PA01) (originally obtained from Prof P.V. Phibbs at the Pseudomonas Genetic Stock Center, East Carolina University) [36] was inoculated into $BR_{prod}$. $CO_2$ produced by the resulting biofilm readily diffused through the highly permeable silicone tube and entered the $BR_{prod}$

annular space, where it was continuously removed by the sweeper gas and carried downstream to Analyzer 1 and beyond. The *P. aeruginosa* biofilm provided a consistent source of CO$_2$ to facilitate the growth and study of the phototrophic biofilm downstream. The bacterial culture was fed with a tryptic soy broth prepared as a 0.6 g/L solution (2% concentration relative to the manufacturer's directions for typical batch cultivation), with a final composition of 0.34 g/L casein peptone, 0.06 g/L soya peptone, 0.05 g/L glucose, 0.1 g/L NaCl, and 0.05 g/L K$_2$HPO$_4$. Media were prepared in distilled water and autoclaved at 121˚C for 20 minutes prior to use.

The phototrophic culture was enriched from an aerated wastewater lagoon in Dundalk, Ontario, Canada [27]. It was cultivated in a modified Bold's Basal Medium (M-BBM) with a composition of 0.22 g/L (NH$_4$)$_2$SO$_4$, 0.025 g/L NaCl, 0.025 g/L CaCl$_2$ · 2H$_2$O, 0.075 g/L MgSO$_4$ · 7H$_2$O, 0.175 g/L KH$_2$PO$_4$, 0.075 g/L K$_2$HPO$_4$, 8.34 mg/L FeSO$_4$. Notably, no carbon was provided in the liquid medium. Illumination was provided continuously via the same fluorescent plant growth light described above. This culture was transferred to fresh medium biweekly, ensuring that the culture used to inoculate BR$_{cons}$-BR$_{cons2}$ in each experiment was not more than 14 days old. Unless otherwise stated, the results reported here are from biofilms at least twenty hours old.

A confocal laser scanning microscope (CLSM) was used to confirm the non-axenic nature of the phototrophic culture. A simple two-channel procedure, described by Lawrence et al. [37] and used subsequently by Barranguet et al. [38] and others, was used to confirm and differentiate the presence of algal and non-algal fractions within the culture used for inoculation of BR$_{cons}$-BR$_{cons2}$. The former was visualized based on chlorophyll autofluorescence, while the latter was distinguished using SYTO 9 green fluorescent nucleic acid stain (S34854; Thermo-Fisher Scientific, Waltham, MA, USA) [37].

A well-mixed 100 µL undiluted culture sample was vacuum filtered through a black polycarbonate filter (0.2 µm pore size, GTBP02500; ThermoFisher Scientific, Waltham, MA, USA). The filter was then immersed in SYTO 9 stain (20 mg/ml) for ten minutes in the dark, before being rinsed twice with deionized water to remove excess stain. The filter was placed on a glass microscope slide and visualized using a Nikon Eclipse 80i-C1 confocal laser scanning microscope (Nikon Instruments Inc., Melville, NY). SYTO 9-stained bacteria were excited using a 488 nm laser and visualized through a 515/530 nm filter, producing a green signal. Microalgal cells conversely, were excited using a 632 nm laser and visualized through a 650 nm long pass filter, producing a red signal.

## Assessing the prevalence of autotrophic-heterotrophic toggling

The phototrophic culture was inoculated into BR$_{cons}$-BR$_{cons2}$ and initially fed M-BBM. CO$_2$ was provided continuously via the heterotrophic biofilm in BR$_{prod}$. After 20 hours, the M-BBM was amended with the labile organic carbon sources glucose (0.25 g/L) and sodium acetate (0.28 g/L). After approximately 30 hours of organic carbon availability, the medium was returned to the original M-BBM composition, which lacked the added carbon sources. Illumination was provided continuously throughout the experiment.

## Investigating the longitudinal arrangement of autotrophic and heterotrophic metabolism

A similar experiment was performed in which the phototrophic culture was inoculated into BR$_{cons}$-BR$_{cons2}$ and initially fed M-BBM, with CO$_2$ provided continuously via the heterotrophic biofilm in BR$_{prod}$. After 30 hours, the M-BBM medium was again amended with glucose and sodium acetate for the remainder of the experiment. In this case however, these organic carbon sources were added at half the concentration used previously (now 0.125 g/L and 0.14

g/L respectively). Illumination was provided by the fluorescent plant growth lights for the duration of the experiment. The CO$_2$ flux within the proximal half (BR$_{cons}$) and distal half of the biofilm (BR$_{cons2}$) are presented separately.

A series of modified light-dark shift tests [27, 39] was performed on BR$_{cons2}$ during the period of organic carbon availability in order to confirm that CO$_2$ uptake observed in this portion of the biofilm was indeed the result of photoautotrophic activity. As in the experiment described above, the biofilm was initially grown under strictly autotrophic conditions, before amending the M-BBM with glucose (0.125 g/L) and sodium acetate (0.14 g/L). BR$_{cons2}$ was then covered with an opaque, light-blocking sheet for two periods of 1.5 hours and 1 hour respectively, with a total of 2.5 hours allowed to elapse between the two dark periods.

### Assessing reversibility in the longitudinal arrangement of autotrophic and heterotrophic metabolism

As in the experiments described above, the phototrophic culture was grown in the CSMS under initial autotrophic conditions, with M-BBM as the liquid growth medium and CO$_2$ provided by the heterotrophic biofilm in BR$_{prod}$. After 38 hours, the M-BBM was once again amended with glucose (0.125 g/L) and sodium acetate (0.14 g/L). However, at hour 60 (22 hours after organic carbon amendment began), the direction of medium and gas flow through BR$_{cons}$-BR$_{cons2}$ was reversed. In this new orientation, fresh growth medium now entered BR$_{cons2}$ from what was previously the effluent end and exited the system at what was previously the influent end of BR$_{cons}$.

After leaving BR$_{prod}$ and passing through Analyzer 1, the gas stream in this reversed-flow orientation subsequently travelled through BR$_{cons2}$ and BR$_{cons}$ in what was previously the upstream direction, with Analyzer 3 positioned between these two BR modules and Analyzer 2 positioned immediately downstream from them. In this configuration, the CO$_2$ flux within the two BR modules was now calculated using Eqs 4 and 5:

$$BR_{cons} \; CO_2 \; flux = Analyzer \; 2 - Analyzer \; 3 \tag{4}$$

$$BR_{cons2} \; CO_2 \; flux = Analyzer \; 3 - Analyzer \; 1 \tag{5}$$

This experiment was repeated in the same manner (autotrophic growth followed by organic carbon amendment and then flow reversal), however the glucose and sodium acetate were added at half the concentration used previously (now 0.0625 g/L and 0.07 g/L respectively).

### Results and discussion

Confocal laser scanning microscopy was used to distinguish algal and non-algal members of the phototrophic culture. Although the culture was maintained under autotrophic conditions, a heterotrophic bacterial fraction was expected to have persisted, as has been described previously for other wastewater-derived phototrophic cultures (e.g. [9]). Microalgal members were recognized by chlorophyll autofluorescence and appeared red, while non-phototrophic bacteria were distinguished via SYTO 9 nucleic acid stain and appeared green (Fig 3). The appearance of non-overlapping red and green cells indicates the presence of both a photosynthetic (algal) and non-photosynthetic (bacterial) fraction in the culture, confirming that the culture was indeed non-axenic.

Algae-bacteria interactions are ubiquitous in nature and play an important role at the base of the trophic web. While numerous types of interactions between these groups are possible, encompassing both positive and negative effects [40], the natural syntrophy surrounding their O$_2$ and CO$_2$ exchange represents an important link between trophic levels and forms the basis

**Fig 3. CLSM images of the phototrophic culture.** (A) The red channel depicts chlorophyll autofluorescence, indicating the presence of algal cells. (B) The green channel depicts SYTO 9-stained bacterial DNA, indicating the distribution of non-photosynthetic bacteria within the culture. (C) Overlay of the red and green signals depicts both cell types, confirming the non-axenic nature of the phototrophic culture.

of healthy ecosystems. Given the co-evolution of microalgae and bacteria over millions of years, it is rare to find microalgae existing as axenic cultures in nature, without some contribution from non-algal, non-photosynthetic microorganisms [41]. While the specific interactions at play within this culture were not fully elucidated, it was nonetheless unsurprising that it appeared to comprise both an autotrophic and heterotrophic fraction (Fig 3), despite being cultivated under autotrophic conditions. In engineered algal systems, non-photosynthetic bacteria are often viewed as mere contamination, necessitating costly and energy-intensive measures to maintain a sterile environment [40]. Recently, this view has begun to shift amidst a growing recognition that natural microalgal-bacterial consortia can represent a cost-effective, efficient alternative for many biotechnological applications [42].

## Autotrophic-heterotrophic toggling by the phototrophic biofilm

When the phototrophic culture was inoculated into the CSMS and grown initially under autotrophic conditions (i.e. no organic carbon), the resulting biofilm grew autotrophically and exhibited net CO$_2$ uptake, denoted by negative CO$_2$ flux values (Fig 4). When the growth medium was amended with glucose and sodium acetate after 20 hours, the CO$_2$ flux began to trend upward. Within approximately 4 hours, the biofilm had clearly shifted from net-

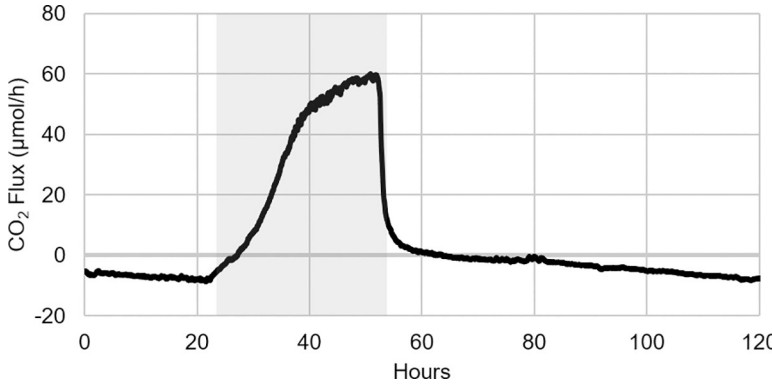

**Fig 4. Biofilm toggling from net-autotrophic to net-heterotrophic growth during organic carbon availability.** Initially, the phototrophic biofilm in BR$_{cons}$-BR$_{cons2}$ grew autotrophically, consuming CO$_2$ provided by BR$_{prod}$ via the sweeper gas. Once organic carbon became available in the culture medium (denoted by the grey box), the biofilm rapidly toggled to CO$_2$-producing net-heterotrophic growth, as indicated by a switch to positive CO$_2$ flux values. When the organic carbon was no longer available in the medium, the biofilm's CO$_2$ production fell dramatically and eventually returned to net-autotrophic growth, as indicated by the return to negative CO$_2$ flux values.

autotrophic to net-heterotrophic growth, denoted by a shift to positive CO$_2$ flux values. During this period of organic carbon availability, CO$_2$ production increased dramatically before beginning to plateau at approximately 60 µmol/h (0.405 µmol/h/cm$^2$).

It should be noted that this CO$_2$ production was a result of the metabolic activity within the phototrophic biofilm only. Given that it was calculated via Eq 1 (Analyzer 3 –Analyzer 1), it does not include any of the CO$_2$ originating from the heterotrophic biofilm in BR$_{prod}$. This meant that the CO$_2$ production observed in the phototrophic biofilm during organic carbon availability could be confidently attributed to the utilization of the two labile organic carbon sources. After 51 hours, the M-BBM was returned to its original composition lacking these organic carbon sources, leading to a rapid decline in the biofilm's CO$_2$ production and an eventual return to net-autotrophic growth. This same CO$_2$ flux behaviour was consistently observed through several experimental repeats.

The data thus supported the underlying hypothesis of this experiment, that a non-axenic phototrophic biofilm readily toggles between net-autotrophic and net-heterotrophic growth, dependent on the availability of labile organic carbon sources. While there is obviously a strong impetus to use wastewater as an inexpensive and abundant growth medium for algal biotechnologies [7, 43], this result speaks to an important consideration in this approach. If bio-sequestration of CO$_2$ (be it from point or diffuse sources) is a primary objective, it may be disadvantageous to grow the sequestering culture in a wastewater stream that is vulnerable to organic nutrient spikes. This is especially relevant in industrial wastewater, or municipal wastewater from plants treating industrial effluents, where wastewater composition can vary widely and transient organic carbon shock loads with concentrations two or more times higher than normal are common [44]. Depending on the robustness and stability of the treatment system, this can result in prolonged periods of elevated effluent BOD, which would impact a downstream algal bio-sequestration system and lead to considerable CO$_2$ production (as seen in Fig 4), an effect which is counter to the overall aim of a sequestration system.

## Longitudinal arrangement of autotrophic and heterotrophic metabolism

The configuration of the CSMS (specifically the placement of Analyzer 2 between BR$_{cons}$ and BR$_{cons2}$), allows for the examination of the respective CO$_2$ flux within the two halves of the phototrophic biofilm. This increased resolution is helpful in revealing metabolic partitioning that may be prevalent in the biofilm. When this approach was applied to the experimental result presented in Fig 4, it was noted that in the presence of the labile organic carbon sources, the vast majority of the biofilm's CO$_2$ production was occurring within the proximal half of the biofilm (Fig 5A). By hour 50, just prior to the end of organic carbon availability, approximately 93% of the observed CO$_2$ production was from BR$_{cons}$, with only about 7% coming from the distal half of the biofilm contained in BR$_{cons2}$. Interestingly, when organic carbon was no longer available in the culture medium, the distal half of the biofilm needed only two hours to return to net-autotrophic growth (CO$_2$ uptake), while the proximal half of the biofilm contained in BR$_{cons}$ took slightly more than two days to return to net-autotrophic growth. The protracted return to autotrophic growth observed in BR$_{cons}$ could have been an EPS effect, whereby some labile organic carbon was retained and subsequently accessible to this portion of the biofilm after organic carbon availability had ceased.

At hour 22 (Fig 5A), when the labile organic carbon sources first became available to the biofilm, there appears to have been enough biomass within BR$_{cons}$ to metabolize and deplete nearly all the glucose and sodium acetate carbon. Very little therefore reached BR$_{cons2}$ where it could be metabolized by the distal half of the biofilm, which accounts for its comparatively low CO$_2$ production. If the organic carbon sources were less abundant (i.e. available in the medium

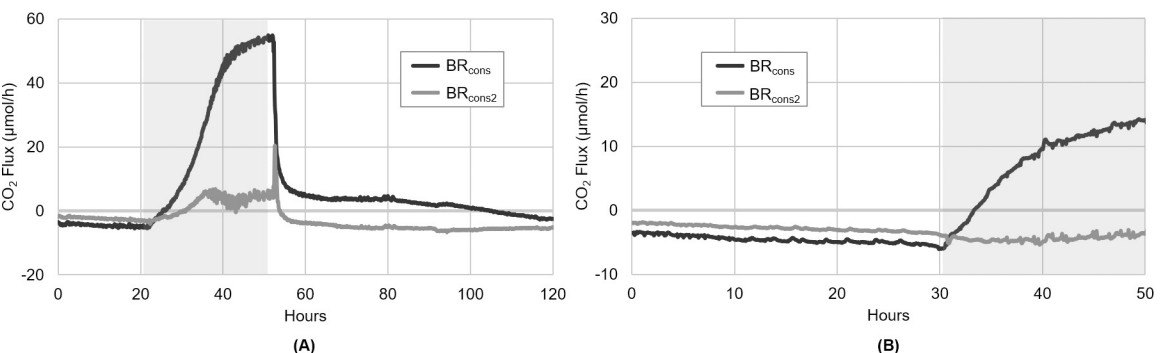

**Fig 5. Proximal and distal biofilm responses during organic carbon availability.** The experiment presented in Fig 4 depicts $CO_2$ flux within the entire linked $BR_{cons}$-$BR_{cons2}$ unit. (A) When examining each half of this biofilm separately, nearly all the observed $CO_2$ production during organic carbon availability (grey box) occurred within $BR_{cons}$, with only very little occurring in $BR_{cons2}$. When organic carbon availability ended, $BR_{cons2}$ quickly returned to $CO_2$-consuming net-autotrophic growth (denoted by a negative $CO_2$ flux), whereas $BR_{cons}$ took over two days to return to $CO_2$-consuming net-autotrophic growth. (B) When the same experiment was performed but with the organic carbon sources provided at half the concentration as compared to Fig 5A, $BR_{cons}$ once again rapidly toggled from $CO_2$-consuming net-autotrophic growth to $CO_2$-producing net-heterotrophic growth during organic carbon availability. However, $BR_{cons2}$ continued to exhibit net-autotrophic growth and $CO_2$ uptake throughout the duration of the experiment, leading to discrete, longitudinally separated regions of net-heterotrophic and net-autotrophic growth occurring simultaneously within the biofilm.

at a lower concentration), one would expect all the glucose and sodium acetate carbon to be depleted within $BR_{cons}$. This would leave the distal half of the biofilm housed in $BR_{cons2}$ without access to any of the supplied organic carbon, causing it to remain in net-autotrophic growth even while the biofilm's proximal half grew heterotrophically. This notion formed the basis of the second hypothesis, which stated that in the presence of labile organic carbon, a phototrophic biofilm of sufficient length exhibits distinct, longitudinally discrete regions of net-heterotrophic and net-autotrophic growth.

To test the hypothesis, a similar experiment was conducted, in which the biofilm was initially grown under autotrophic conditions with $CO_2$ supplied from the heterotrophic biofilm in $BR_{prod}$. The M-BBM fed to $BR_{cons}$-$BR_{cons2}$ was then once again amended with glucose and sodium acetate, however this time at half the concentration used previously (0.125 g/L and 0.14 g/L respectively). Before this organic carbon amendment, both halves of the biofilm were exhibiting autotrophic growth and $CO_2$ uptake, indicated by their negative $CO_2$ flux values (Fig 5B). However, when the labile organic carbon sources became available in the medium after 30 hours, the proximal half of the biofilm ($BR_{cons}$) rapidly shifted from $CO_2$-consuming autotrophic growth to $CO_2$-producing heterotrophic growth. The $CO_2$ flux in the distal half of the biofilm ($BR_{cons2}$) conversely, remained relatively unchanged and continued to exhibit net-autotrophic growth ($CO_2$ uptake) for the entire experiment.

In three subsequent experimental repeats, this same pattern of metabolic partitioning was observed. In the presence of the labile organic carbon sources, the biofilm's proximal half produced $CO_2$, while the biofilm's distal half continued to consume $CO_2$. When $BR_{cons2}$ was placed in the dark for two periods of 1.5 hours and 1 hour respectively during the period of organic availability, a rapid response was observed in which the $CO_2$ flux in this portion of the biofilm shifted from negative ($CO_2$ uptake) to positive ($CO_2$ production) (Fig 6). This dark-induced interruption of photosynthesis thereby confirmed that the $CO_2$ uptake observed in $BR_{cons2}$ during organic carbon availability was in fact attributable to autotrophic activity (as opposed to abiotic leakage or passive diffusion out of the system). The small amount of $CO_2$ production observed during this darkness is likely attributable to underlying baseline algal respiration. When illumination resumed, so too did the photoautotrophic activity in $BR_{cons2}$,

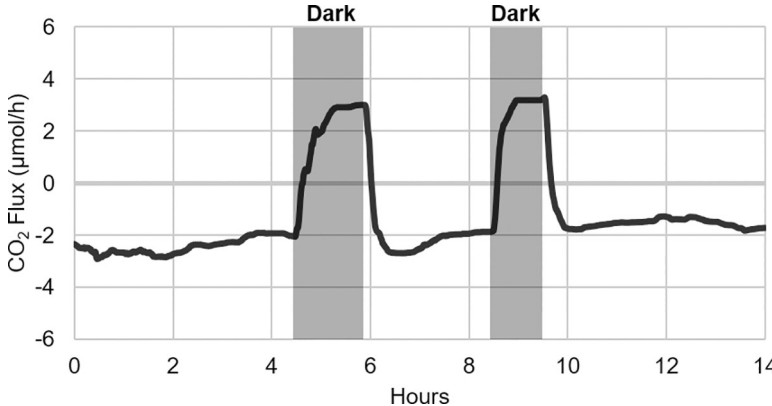

**Fig 6. Dark response in BR$_{cons2}$ during organic carbon availability.** In order to confirm that the CO$_2$ uptake observed in BR$_{cons2}$ during organic carbon availability was indeed attributable to photosynthetic CO$_2$ fixation, BR$_{cons2}$ was placed in the dark twice for at least one hour. Both times, CO$_2$ uptake stopped almost immediately. This signified the rapid cessation of photosynthesis and its related CO$_2$ fixation, and provided confirmation that the phototrophic biofilm was indeed exhibiting distinct, longitudinally discrete regions of net-heterotrophic and net-autotrophic growth when exposed to both inorganic and organic carbon (CO$_2$ as well as glucose and sodium acetate).

causing this portion of the biofilm to quickly rebound to approximately the same negative CO$_2$ flux values seen before the dark test. Given this result, it can be concluded that in the presence of both inorganic and organic carbon (CO$_2$ as well as glucose and sodium acetate), the phototrophic biofilm was in fact exhibiting distinct, longitudinally discrete regions of net-heterotrophic and net-autotrophic growth.

Previous studies have described the use of non-axenic phototrophic cultures for simultaneous inorganic and organic carbon removal. van der Ha et al. [45] for example, co-cultured microalgae and methane-oxidizing bacteria to achieve concurrent microbial methane oxidation and CO$_2$ uptake. Vu and Loh [46] described a symbiotic consortium of *C. vulgaris* and *P. putida*, where the algal fraction grew photoautotrophically using CO$_2$ produced by the bacterium. This in turn eliminated the need for aeration by providing the oxygen needed by the obligate aerobe *P. putida* to carry out efficient glucose biodegradation. A similar co-culture of *P. putida* and *C. vulgaris* described by Mujtaba et al. [47] exhibited the synergistic removal of organic carbon by the former and inorganic nutrients by the latter.

While there is much literature pertaining to microalgal-bacterial symbiosis in co-culture or synthetic consortia, comparatively few studies have examined this behaviour in natural phototrophic consortia within the context of engineered biofilm photobioreactors. Whereas co-cultures still require onerous measures to maintain sterility and prevent contamination, natural non-axenic cultures can offer a level of robustness and functional redundancy that makes contaminating organisms much less of a concern [13, 25]. Although wastewater treatment and CO$_2$ sequestration have been studied separately for many years, approaches for effectively linking these via non-axenic phototrophic biofilms are lacking [8].

Microalgal biofilm growth systems fall into two broad categories: those in which the biofilm is mobile (via movement of the attachment material), and those in which the biofilm remains stationary (no movement of the attachment material) [48]. Mobile systems include for example rotating algal biofilm reactors (RABRs), in which the biofilm is attached to a solid or fibrous support material coiled around a rotating cylindrical drum that alternatingly exposes cells to the liquid and gas phases. Such systems have been shown to achieve high inorganic nutrient removal and biomass production when operating within a raceway pond [23]. However, the energy required for their moving parts can be a complicating factor. In stationary

growth systems conversely, biofilms form on a fixed attachment material with liquid growth medium flowing over the surface of the biofilm. Tubular photobioreactors for example, are closed, stationary systems which enable significant control over conditions and offer high surface to volume ratios [49].

Insight generated by the CSMS can be useful in informing the design and operation of similar scaled-up tubular systems. This can include for example determining what fraction of total reactor length is required to deplete nutrients. Since this length will vary with key factors such as influent nutrient load and flow rate, the utility of the CSMS lies in its innovative approach to informing such optimization. The longitudinal arrangement of discrete heterotrophic and autotrophic regions within the phototrophic biofilm (as described in Fig 5), suggests that significant simultaneous removal of organic and inorganic carbon may be possible in large-scale tubular photobioreactors using non-axenic phototrophic biofilms. Ostensibly, the former could be metabolized in the proximal portion of the biofilm through heterotrophic metabolism, with the latter being taken up and fixed through autotrophic metabolism occurring in the distal portion of the biofilm, where assimilation of inorganic nitrogen and phosphorus nutrients would also take place.

Despite the potential benefits offered by tubular photobioreactors in terms of CO$_2$ capture and contaminant removal, the energy required for operation, as well as the need for cleaning and maintenance, remain barriers limiting their widespread use. As such, there is still a need for further research in this area, focused on expanding our understanding of biofilm behaviour within these systems and developing strategies for improving their overall cost-effectiveness.

## Reversibility in the longitudinal arrangement of autotrophic and heterotrophic metabolism

In the presence of inorganic and organic carbon, the phototrophic biofilm consistently exhibited longitudinally discrete regions of net-heterotrophic and net-autotrophic metabolism, with the location of these regions dictated by the gradient of organic carbon availability resulting from heterotrophic metabolism in the proximal portion of the biofilm. It was hypothesized that these discrete regions of net-autotrophic and net-heterotrophic growth are therefore transient and reversible based on the direction of nutrient flow and availability. That is, when the direction of flow is reversed, the locations of the autotrophic and heterotrophic regions will flip.

To test this hypothesis, a similar experiment was performed in which the biofilm initially grew autotrophically with CO$_2$ as the sole carbon source. When glucose and sodium acetate were added to the growth medium, BR$_{cons}$ once again toggled to CO$_2$-producing heterotrophy, while BR$_{cons2}$ continued to exhibit net-autotrophic CO$_2$ uptake (Fig 7A). At hour 60 however, the direction of medium and gas flow through these two BR modules was reversed, such that BR$_{cons2}$ now contained the "proximal" half of the biofilm, gaining access to the fresh, un-depleted organic carbon sources provided in the medium. BR$_{cons}$ conversely, was now downstream of BR$_{cons2}$ and represented the "distal" half of the biofilm.

After the direction of flow was reversed, a sharp increase in CO$_2$ flux and a rapid switch from autotrophic to heterotrophic growth was observed in BR$_{cons2}$ (Fig 7A). This aligned with expectations, given that this portion of the biofilm was now first to encounter and metabolize the organic carbon sources. In previous experiments (and before reversing the direction of flow in this experiment), this portion of the biofilm was only able to access what little (if any) of the organic carbon could pass through BR$_{cons}$ un-metabolized.

It is notable that the slope in the BR$_{cons2}$ CO$_2$ flux after hour 60 is remarkably similar to the slope in BR$_{cons}$ after hour 38 (Fig 7A), suggesting that both halves of the biofilm exhibited the same metabolic response (and started producing CO$_2$ at nearly identical rates), when that

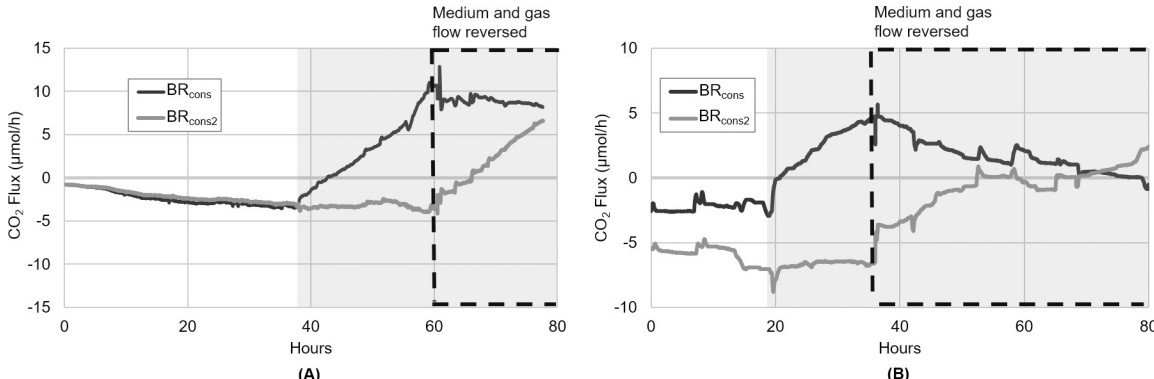

**Fig 7. Proximal and distal biofilm responses to reverse flow direction during organic carbon availability.** (A) The phototrophic biofilm initially grew autotrophically using $CO_2$ from $BR_{prod}$. As previously observed, $BR_{cons}$ shifted to $CO_2$-producing net-heterotrophic growth after organic carbon amendment (grey box), while $BR_{cons2}$ continued in $CO_2$-consuming net-autotrophic growth. When the direction of medium and gas flow through $BR_{cons}$-$BR_{cons2}$ was reversed at hour 60, the $CO_2$ flux in $BR_{cons}$ stopped increasing and gradually decreased. The $CO_2$ flux in $BR_{cons2}$ conversely, increased rapidly and began exhibiting net $CO_2$ production. (B) This experiment was repeated, but with the glucose and sodium acetate concentrations decreased by half. Once again, both halves of the biofilm grew autotrophically initially, before $BR_{cons}$ toggled to $CO_2$-producing heterotrophy after organic carbon amendment (grey box). When the direction of medium and gas flow in $BR_{cons}$-$BR_{cons2}$ was reversed, $CO_2$ production in $BR_{cons}$ stopped increasing and began to decrease, ultimately crossing zero just prior to the end of the experiment. $BR_{cons2}$ conversely gradually stopped consuming $CO_2$ after flow direction was reversed, eventually crossing over to net $CO_2$ production.

portion of the biofilm was the first to encounter the organic carbon sources. This result speaks to the significant analytical utility of the CSMS. The ability to visualize the biofilm's rate of $CO_2$ consumption and/or production in real-time means that growth conditions can be fine-tuned to ensure that these rates are effectively optimized. Increasing the number of $CO_2$ analyzers positioned along the length of the biofilm would also enable greater longitudinal resolution, which can further inform optimal nutrient loading and reactor length under a given set of conditions.

When $BR_{cons}$ became the distal end of the biofilm and was no longer receiving the fresh, un-depleted organic carbon sources, its rate of $CO_2$ production began to gradually decrease (Fig 7A). However, this portion of the biofilm did not indicate an overall shift to net-autotrophic growth as expected. As mentioned previously, EPS may have played a role in this muted response by retaining stores of organic carbon from earlier in the experiment, which could then be taken up and metabolized after the direction of flow was reversed. Interestingly, in natural, non-axenic phototropic biofilms, much of the initial EPS originates from heterotrophic bacteria [50]. It is also plausible that following flow reversal at hour 60, the portion of the biofilm in $BR_{cons2}$ was not yet dense enough to fully deplete the organic carbon it now encountered, therefore leaving some available for the "distal" half of the biofilm now housed in $BR_{cons}$. Although the precise explanation necessitates additional biofilm analyses which are ultimately beyond the scope of this paper, it was postulated that in either case these effects would be minimized, and the $BR_{cons}$ $CO_2$ flux after flow reversal would eventually cross zero and exhibit $CO_2$ uptake, if the organic carbon sources were available in the medium at a lower concentration.

To test this, the experiment was repeated but with the glucose and sodium acetate supplied at half the concentration used previously (now 0.0625 g/L and 0.07 g/L respectively). This experiment was also allowed to run for a longer period following the reversal of flow direction. $BR_{cons2}$ once again exhibited an increase in its $CO_2$ flux values after flow reversal, eventually achieving net $CO_2$ production (Fig 7B). As predicted, $BR_{cons}$ in this case showed a steady decline in $CO_2$ flux after flow reversal, reaching zero by approximately hour 70.

Given that the $CO_2$ flux in both $BR_{cons}$ and $BR_{cons2}$ trended in opposite directions after flow reversal (Fig 7B), and considering the outcome of the previous experiment (Fig 7A), the upstream and downstream partitioning of net-heterotrophic and net-autotrophic metabolism appears to be reversible, with the position of these regions dictated by nutrient flow direction and hence availability, thus supporting the third hypothesis. Although a comprehensive community composition analysis would be needed to make definitive assertions, these results suggest that this rapid metabolic partitioning is likely not the result of significant changes in biofilm member composition, but rather a functional redundancy within the culture and changes in the metabolic activity of members already present before the influx of organic carbon.

The experiments described here present an elegant approach to the study of phototrophic biofilms in order to gain important insight regarding their behaviour under changing conditions. Such information can inform new and relevant research questions and contribute to efforts aimed at scaling and industrializing algal growth systems, where the ability to understand, predict, and optimize biofilm growth and activity is critical. In integrated wastewater treatment-$CO_2$ sequestration systems utilizing non-axenic phototrophic biofilms, the balance between organic and inorganic carbon metabolism is a major factor that dictates the biofilm's overall $CO_2$ flux. In this study, the phototrophic biofilm exhibited a rapid response and began growing heterotrophically when exposed to organic carbon, reaching a $CO_2$ production rate of 60 μmol/h after approximately 30 hours. This suggests that non-axenic phototrophic biofilms are indeed able to readily toggle between net-autotrophic $CO_2$ capture and net-heterotrophic $CO_2$ production based on the availability of labile organic carbon sources. When the organic carbon sources were less abundant (i.e. provided at a much lower concentration), the biofilm took only 4 hours to exhibit longitudinally discrete regions of autotrophic and heterotrophic growth in the proximal and distal portions of the biofilm respectively. The apparent reversibility in the biofilm's response to changing carbon conditions suggests a robustness and versatility within non-axenic phototrophic biofilms which may be exploitable in engineered phototrophic biofilm systems to achieve simultaneous organic and inorganic carbon removal.

Hypothesis-driven research aimed at generating fundamental insights about phototrophic biofilms are important as the need for effective $CO_2$ management and mitigation strategies increases, and the motivation for linking these to enhanced wastewater treatment grows. The study presented here therefore should represent a key step forward in this endeavour.

## Acknowledgments

The authors would like to acknowledge Jamie Nguyen for her help with lab work that contributed to this article.

## Author Contributions

**Conceptualization:** Patrick Ronan, Otini Kroukamp.

**Data curation:** Patrick Ronan.

**Formal analysis:** Patrick Ronan.

**Investigation:** Patrick Ronan.

**Methodology:** Patrick Ronan.

**Resources:** Steven N. Liss, Gideon Wolfaardt.

**Supervision:** Otini Kroukamp, Steven N. Liss, Gideon Wolfaardt.

**Writing – original draft:** Patrick Ronan.

**Writing – review & editing:** Otini Kroukamp, Steven N. Liss, Gideon Wolfaardt.

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
