## [Decision Letter · Decision Letter 0]

5 May 2021

PONE-D-21-12369

Interaction between CO2-consuming autotrophy and CO2-producing heterotrophy in non-axenic phototrophic biofilms

PLOS ONE

Dear Dr. Wolfaardt,

Thank you for submitting your manuscript to PLOS ONE. After careful consideration, we feel that it has merit but does not fully meet PLOS ONE’s publication criteria as it currently stands. Therefore, we invite you to submit a revised version of the manuscript that addresses the points raised during the review process.

We look forward to receiving your revised manuscript.

Kind regards,

Martin Koller, Ph.D.

Academic Editor

PLOS ONE

Journal Requirements:

3. In your Methods section, please provide additional details regarding the cell lines used in your study and ensure you have described the source. For more information regarding PLOS' policy on materials sharing and reporting, see https://journals.plos.org/plosone/s/materials-and-software-sharing#loc-sharing-materials, and for more information on PLOS ONE's guidelines for research using cell lines, see https://journals.plos.org/plosone/s/submission-guidelines#loc-cell-lines.

Reviewers' comments:

Reviewer's Responses to Questions

**Comments to the Author**

1. Is the manuscript technically sound, and do the data support the conclusions?

Reviewer #1: Yes

Reviewer #2: Yes

2. Has the statistical analysis been performed appropriately and rigorously? 

Reviewer #1: N/A

Reviewer #2: I Don't Know

3. Have the authors made all data underlying the findings in their manuscript fully available?

Reviewer #1: Yes

Reviewer #2: Yes

4. Is the manuscript presented in an intelligible fashion and written in standard English?

Reviewer #1: Yes

Reviewer #2: Yes

5. Review Comments to the Author

Reviewer #1: General comments

The authors have investigated the complex interactions of non-axenic biofilms consisting of photo-autotrophic and heterotrophic microorganisms and experimentally investigate on kinetics and dynamics of microbial growth in reaction to changing feed and gas compositions.

The CO2 source was exhaust gas of a growing culture of Pseudomonas sp. cultivated in a tubular reactor, in a similar setup as for the phototrophic biofilm. At this point, no explanation was given why such a culture and not a standard gas mixture containing controllable concentration of CO2 was chosen. The statement provided in the lines 470ff can and should be moved to the methods section.

The experimental setup reads comparably simple while the operation modes have some complexity in their variation of illumination, mineral nutrient and organic substrate supply with or without reverting medium flow direction.

The photobioreactor was inoculated with a mixed culture of phototrophic microbes isolated and enriched from a local surface water body. Although it is probably impossible to identify all involved species in the mixture, at least normal light microscopy photographs should be provided from the inoculum – or from the first established biofilm and from experimental stages at the times of operation changes. Such a visual documentation, as simple as it is, would improve the readers imagination about the ongoing population changes during different operation modes. If not available – a pity, btw. – some some other wordly descriptions should be provided. The one status photo in Fig.3. is a good example, but does not allow to follow a highly dynamic system over the course of roughly 3 to 5 days.

By monitoring CO2 consumption and CO2 evolution as indicators of either heterotrophic or autotrophic metabolic dominance in the biofilm, some general and valuable information with praxis relevance about microalgae production from wastewater is provided. A natural day-night cycle should be considered, as a real scale algae production utilising wastewater nutrients will not be artificially illuminated. During the dark hours also the phototrophic biomass will consume oxygen and will release CO2, interfering with the overall net carbon balance. Maybe such extrapolations are possible from the experiment sequences with artificial shadowing of the bioreactor.

Specific comments

Introduction lines 73 - 75: While microalgal biomass can indeed be used as fertilizer, feed and food additive, any biomass grown on waste has a restricted application range – a fact to be considered.

Discussion, lines 296 – 306: While all this information is correct and valuable, it is, nevertheless, of theoretical nature and not a direct outcome of experimental results.

Discussion, lines 454 – 457: The recommendation of closed tubular photobiorectors for microalgae production from wastewater is highly theoretical. Unpredictable population development and the need for cleaning and maintenance do, by far, outweigh the potential benefits of a technically controlled environment. Flow resistance in the tubes requires way more pumping energy than a raceway pond the authors criticize some sentences above (lines 451 ff).

Discussion, lines 478ff (reversibility): It‘s a pity that no information is provided about biofilm development and biofilm composition. It seems to be of major importance to understand if the reaction to the availability of organic carbon in the medium is caused by metabolic changes of a static biofilm composition or by a change of the ratio of photo- to heterotrophic cells (reactive growth). This can be important to know as a stable biofilm can be operated over a longer time while a reactively changing biofilm will cause problems in a large scale plant. However, the authors mention in lines 529ff that such analysis were beyond their possibilities for this manuscript.

Reviewer #2: The manuscript is based on a very interesting hypothesis , and undertaken in a suitable manner.

The Introduction, upto line no 99, is too wordy and seems to be unnecessary. The Last few figures can be combined and presented in a better manner.

I would appreciate a bit more specific data/results in the abstract and conclusions, to make it more scientific.

6. PLOS authors have the option to publish the peer review history of their article (what does this mean?). If published, this will include your full peer review and any attached files.

Reviewer #1: **Yes: **Ines Fritz

Reviewer #2: No

---

## [Author Response · Author response to Decision Letter 0]

28 May 2021

Please also see uploaded document "Response to reviewers"; which is exactly the same as the text below

Reviewer #1: General comments

The authors have investigated the complex interactions of non-axenic biofilms consisting of photo-autotrophic and heterotrophic microorganisms and experimentally investigate on kinetics and dynamics of microbial growth in reaction to changing feed and gas compositions.

The CO2 source was exhaust gas of a growing culture of Pseudomonas sp. cultivated in a tubular reactor, in a similar setup as for the phototrophic biofilm. At this point, no explanation was given why such a culture and not a standard gas mixture containing controllable concentration of CO2 was chosen. 

We opted for this approach as it highlights the utility of the CSMS biofilm reactor modules, wherein the permeability of the inner silicone tube can facilitate not only the delivery of CO2 to a biofilm, but also the collection and subsequent shuttling of CO2 out of a biofilm (this is mentioned on line 132). Although not a primary focus of the present study, this is a noteworthy property which could be used in initiatives aimed at collecting and subsequently sequestering CO2 emitted by real-world CO2-producing biological processes.

We have added the passage below at line 197 to expand on the rationale behind the CO2 source used in this study (note that line numbers correspond to the revised, unmarked version of the manuscript).

In this study, CO2 was provided to the phototrophic biofilm via the heterotrophic bacterial biofilm. At steady state, the biofilm’s consistent CO2 output serves to demonstrate the supply of an inexpensive and renewable CO2 source to facilitate the growth and subsequent study of the phototrophic biofilm downstream. While it would also be feasible to accomplish this using a gas tank with a known or controllable CO2 concentration, this approach highlights the utility in the design of the CSMS biofilm reactor modules, where the permeability of the inner silicone tube can facilitate not only the delivery of CO2 to a biofilm, but also the collection and subsequent shuttling of CO2 out of a biofilm.

The statement provided in the lines 470ff can and should be moved to the methods section.

Thank you for pointing this out. We have adjusted the text accordingly by moving the passage in question from the Discussion to the Materials and Methods (now covered in lines 196-209). 

The experimental setup reads comparably simple while the operation modes have some complexity in their variation of illumination, mineral nutrient and organic substrate supply with or without reverting medium flow direction.

The photobioreactor was inoculated with a mixed culture of phototrophic microbes isolated and enriched from a local surface water body. Although it is probably impossible to identify all involved species in the mixture, at least normal light microscopy photographs should be provided from the inoculum – or from the first established biofilm and from experimental stages at the times of operation changes. Such a visual documentation, as simple as it is, would improve the readers imagination about the ongoing population changes during different operation modes. If not available – a pity, btw. – some other wordly descriptions should be provided. The one status photo in Fig.3. is a good example, but does not allow to follow a highly dynamic system over the course of roughly 3 to 5 days.

We agree that the tracking of microbial community changes over time would be a good compliment to the story presented in the paper. However, such analyses were beyond the scope of this study, due in part to the fact that in the current iteration of the CSMS, the biofilm is housed within two layers of tubing, making time-resolved in situ biofilm imaging unfeasible. Unfortunately, this tube-within-a-tube design feature is critical to the system’s real-time CO2 monitoring capability. Instead of a microbial composition focus, we therefore opted for a “functional” focus, tracking and describing the changes in community function rather than community composition. With this focus, our system was able to speak more to metabolic behaviour (i.e. the “readiness” of a certain microbial fraction (e.g. heterotrophs) to act upon a change in their environment within a certain time frame) rather than population changes. The biofilm’s ability to toggle its CO2 flux in mere minutes suggests that it is indeed changes in metabolic behaviour which is likely predominating in the period immediately following feed amendment. We acknowledge that this will most probably be accompanied by eventual population changes, though one would expect the metabolic impact of these changes to manifest on a comparatively longer timescale, and that these changes will be relatively small due to the functional redundancy inherent to microbial communities. In light of this, we feel that despite the lack of time-resolved biofilm imaging, the study’s experimental design is sound, and the insight generated still represents a meaningful and useful contribution to this area of study. 

By monitoring CO2 consumption and CO2 evolution as indicators of either heterotrophic or autotrophic metabolic dominance in the biofilm, some general and valuable information with praxis relevance about microalgae production from wastewater is provided. 

A natural day-night cycle should be considered, as a real scale algae production utilising wastewater nutrients will not be artificially illuminated. During the dark hours also the phototrophic biomass will consume oxygen and will release CO2, interfering with the overall net carbon balance. Maybe such extrapolations are possible from the experiment sequences with artificial shadowing of the bioreactor.

We agree that day-night cycles are an important consideration in algae production from wastewater. It is true that CO2 will be produced in the dark, and indeed some baseline algal CO2 production will continue to occur even in the light. Our reported CO2 flux data encompasses all ongoing CO2-consuming and CO2-producing processes taking place in the biofilm, under lighting conditions which actively and continuously promote photosynthesis and related CO2 uptake (as CO2 sequestration was a primary focus of the paper). We acknowledge that light/dark cycling will have an impact on this carbon balance, as the reviewer indicated, and therefore feel that this study provides a useful basis of comparison for future studies aimed at exploring this impact and ultimately optimizing light/dark cycling to maximize specific beneficial parameters (e.g. CO2 uptake, biomass production, wastewater nutrient removal etc.).

Specific comments

Introduction lines 73 - 75: While microalgal biomass can indeed be used as fertilizer, feed and food additive, any biomass grown on waste has a restricted application range – a fact to be considered.

Thank you for highlighting this important consideration. We have added the statement below at line 76 to address this point. 

Although the range of allowable applications for waste-grown biomass remains somewhat restricted [16], further research and evidence-based policymaking focused on risk mitigation could lead to a paradigm in which microalgal CO2-sequestration, enhanced wastewater treatment, and biomass generation may be effectively combined [7, 17, 18].

Discussion, lines 296 – 306: While all this information is correct and valuable, it is, nevertheless, of theoretical nature and not a direct outcome of experimental results.

It was not our intention to present this brief discussion of algal-bacterial interactions as a direct outcome of our experiment, but rather we include this to provide some helpful context surrounding the observed presence of non-autotrophic members in our phototrophic culture. We have added the sentence below at line 318 in order to clarify.

While the specific interactions at play within this were not fully elucidated, it was nonetheless unsurprising that it appeared to comprise both an autotrophic and heterotrophic fraction (Fig 3), despite being cultivated under autotrophic conditions.

Discussion, lines 454 – 457: The recommendation of closed tubular photobiorectors for microalgae production from wastewater is highly theoretical. Unpredictable population development and the need for cleaning and maintenance do, by far, outweigh the potential benefits of a technically controlled environment. Flow resistance in the tubes requires way more pumping energy than a raceway pond the authors criticize some sentences above (lines 451 ff).

Thank you for highlighting this point. Open systems (e.g. raceways) and closed systems (e.g. tubular PBRs) certainly both possess an array of advantages and disadvantages, which we accept when stacked against each other relegates the latter to a more “theoretical” nature. As pointed out, the need for pumping energy and cleaning in tubular systems are important factors which, among other considerations, currently limit their widespread use. We have added the sentence below at line 488 to note this. 

Despite the potential benefits offered by tubular photobioreactors in terms of CO2 capture and contaminant removal, the energy required for operation, as well as the need for cleaning and maintenance, remain barriers limiting their widespread use. As such, there exists a need for further research in this area, focused on expanding our understanding of biofilm behaviour within these systems and developing strategies for improving the overall cost-effectiveness of their operation.

Discussion, lines 478ff (reversibility): It’s a pity that no information is provided about biofilm development and biofilm composition. It seems to be of major importance to understand if the reaction to the availability of organic carbon in the medium is caused by metabolic changes of a static biofilm composition or by a change of the ratio of photo- to heterotrophic cells (reactive growth). This can be important to know as a stable biofilm can be operated over a longer time while a reactively changing biofilm will cause problems in a large scale plant. However, the authors mention in lines 529ff that such analysis were beyond their possibilities for this manuscript.

In this study, we documented the phototrophic biofilm’s interesting response to organic carbon availability, in which longitudinally discrete regions of net CO2 production and net CO2 consumption appear in the proximal and distal halves of the biofilm, respectively. As pointed out by the reviewer, an interesting question arises as to whether the observed response is attributable to metabolic changes within a static biofilm composition or changes in the ratio of photo- to heterotrophic cells. While a definitive answer was indeed beyond the scope of this paper, we suspect that over time both scenarios may be at play, but with the impact of each manifesting on different timescales. The biofilm’s near immediate toggling of its CO2 flux in response to organic carbon suggests that a metabolic change within existing community members likely predominates initially. It is plausible that such a metabolic shift will then eventually lead to a community composition response, which may well include a changing ratio of photo- to heterotrophic members. However, given typical microbial growth rates, we would expect this response and its related impact on overall biofilm CO2 flux to materialize on a comparatively longer timescale than the metabolic response. Nevertheless, we agree that this question is worthy of further examination in the future, as it may have implications for the long-term operation of such biofilm systems.

Reviewer #2 General Comments 

The manuscript is based on a very interesting hypothesis, and undertaken in a suitable manner.

The Introduction, up to line no 99, is too wordy and seems to be unnecessary. 

In the Introduction, we have strived to clearly present the motivation for studying algae in general, as well the rationale behind this study in particular, which we feel provides the reader with important context. Since algae can offer significant benefit in terms of CO2 sequestration, wastewater treatment, and biomass production, we felt it necessary to briefly describe these three applications and highlight the potential for a future paradigm in which they may be effectively combined. 

We accept the reviewer’s opinion regarding the wordiness in the Introduction and have taken measures to cut down on this in several areas (e.g. lines 26, 27, 33, 51, 52, 56, 72, 80). 

The Last few figures can be combined and presented in a better manner.

We strive to present our data in the clearest and most comprehendible manner. The last two figures each include two different data sets, both of which depict a change in feed composition as well as a subsequent reversal of flow direction. As such, we are wary of trying to convey too much information in a single graph, which could ultimately detract from its clarity. However, since these graphs present very similar experiments which are both speaking to the same hypothesis, we do agree that it would be logical to present them in a more cohesive manner. We have therefore combined what was previously Fig 7 and Fig 8 into a single Fig 7, with the two graphs labelled as “7A” and “7B”, as we have done previously with Fig 5. The caption and references to this figure have been adjusted in the text to reflect this change. 

I would appreciate a bit more specific data/results in the abstract and conclusions, to make it more scientific.

Thank you for this recommendation. We have made several adjustments to the text in the Abstract (starting line 33) as well as in the concluding passage (starting at line 577) in order to provide more specificity when highlighting the results.

---

## [Editor Report · Decision Letter 1]

1 Jun 2021

Interaction between CO2-consuming autotrophy and CO2-producing heterotrophy in non-axenic phototrophic biofilms

PONE-D-21-12369R1

Dear Dr. Wolfaardt,

We’re pleased to inform you that your manuscript has been judged scientifically suitable for publication and will be formally accepted for publication once it meets all outstanding technical requirements.

Kind regards,

Martin Koller, Ph.D.

Academic Editor

PLOS ONE
---

## [Editor Report · Acceptance letter]

7 Jun 2021

PONE-D-21-12369R1 

Interaction between CO_2_-consuming autotrophy and CO_2_-producing heterotrophy in non-axenic phototrophic biofilms 

Dear Dr. Wolfaardt:

I'm pleased to inform you that your manuscript has been deemed suitable for publication in PLOS ONE. Congratulations! Your manuscript is now with our production department. 

Kind regards, 

on behalf of

Dr. Martin Koller 

Academic Editor

PLOS ONE